# COVID-19 Racism and Chinese American Families’ Mental Health: A Comparison between 2020 and 2021

**DOI:** 10.3390/ijerph20085437

**Published:** 2023-04-07

**Authors:** Charissa S. L. Cheah, Huiguang Ren, Xiaoli Zong, Cixin Wang

**Affiliations:** 1Department of Psychology, University of Maryland, Baltimore County, Baltimore, MD 21250, USA; 2Department of Counseling Psychology, Higher Education and Special Education, University of Maryland, College Park, MD 20742, USA

**Keywords:** multiple forms of racism, COVID-19, cumulative impacts, Chinese American parents and youth, mental health

## Abstract

This study compared rates of multiple forms of COVID-19 racism-related discrimination experiences, fear/worries, and their associations with mental health indices among Chinese American parents and youth between 2020 and 2021. Chinese American parents of 4- to 18-year-old children and a subsample of their 10- to 18-year-old adolescents completed surveys in 2020 and 2021. A high percentage of Chinese American parents and their children continued to experience or witness anti-Chinese/Asian racism both online and in person in 2021. Parents and youth experienced less vicarious discrimination in person but more direct discrimination (both online and in person) and reported poorer mental health in 2021 than in 2020. Associations with mental health were stronger in 2021 than in 2020 for parents’ and/or youth’s vicarious discrimination experiences, perceptions of Sinophobia, and government-related worries, but weaker only for parents’ direct discrimination experiences. The spillover effect from parents’ vicarious discrimination experiences and Sinophobia perceptions to all youth mental health indices were stronger in 2021 than in 2020. Chinese American families experienced high rates of racial discrimination across multiple dimensions, and the detrimental impacts on their mental health were still salient in the second year of the pandemic. Vicarious and collective racism may have even stronger negative impacts on mental health and well-being later in the pandemic. Decreasing health disparities for Chinese Americans and other communities of color requires extensive, long-term national efforts to eliminate structural aspects of racism.

## 1. Introduction

Anti-Asian racism and xenophobia have surged in the United States since the outbreak of the COVID-19 pandemic in December 2019 and continued to rise in 2021. Between March 2020 and 2022, the Stop AAPI (Asian American and Pacific Islanders) Hate Reporting Center received 11,467 reports of racism and discrimination against Asian Americans in the United States, with Chinese Americans comprising the majority (43%) of the adult respondents and 9% of the youth respondents (below 18 years old) [1]. Unfortunately, the rate of anti-Asian incidents increased by 164% from 2020 to 2021 [2], indicating no abatement in reports one year after the pandemic outbreak.

The harmful effects of COVID-19-related racial discrimination on Asian Americans’ mental health have been documented in several studies conducted shortly after the pandemic outbreak. These findings consistently show that Asian Americans experienced racial discrimination during the pandemic [3,4,5,6,7], which has been associated with negative mental health outcomes, such as greater anxiety [3,4], more depressive symptoms [4,6], and poorer psychological well-being [5,7]. Although most of this research focused on experiences of racial discrimination and the negative impact of these experiences on adults of Asian heritage [8,9], the negative mental health impacts of COVID-19-related racial discrimination are likely to be cumulative, especially for youth [10]. Cheah et al. found that nearly half of Chinese American parents and youth reported being directly targeted by COVID-19-related racial discrimination and more than two-thirds reported vicarious racial discrimination online and/or in person [7]. In addition, both parents (49%) and youth (71%) perceived health-related Sinophobia in America, which was thought to be perpetuated by the media (parents: 50%; youth: 56%) [7], as people who use social media frequently are more likely to view Chinese as a symbolic threat [11]. Although previous epidemics (e.g., SARS, H1N1, MERS, Zika, Ebola) also resulted in stigma and xenophobia towards various subgroups from specific regions [12], the current geopolitical tension between the United States and China, and the intermingling among health, race, and national statuses of Chinese Americans may have intensified Sinophobia and its negative impact on Chinese Americans [13]. Higher levels of parent- and youth-perceived racism and racial discrimination were associated with numerous indices of poorer mental health [7,10,14].

COVID-19-related experiences of discrimination, Sinophobia, and racism-related fear/worries can affect children’s mental health both directly and indirectly through their parents’ experiences [7]. For both children and their parents, the consequences of being exposed to the chronic stressor of COVID-19-related racial discrimination may not fully manifest in the early part of the pandemic [15]. According to minority stress theory [16], accumulative experiences of racial discrimination can contribute to poor physical and mental health of minoritized populations in the long-term through physiological mechanisms related to stress response (i.e., allostatic overload) [17]. Indeed, about two-thirds of Asian Americans who experienced racial trauma (i.e., psychological or emotional harm caused by racism) in November 2020 still met the criteria in January to March 2021 [18]. Asian American youth’s COVID-19-related discrimination was positively associated with symptoms of posttraumatic stress disorder [19].

Racism occurs across various interpersonal, institutional, and cultural contexts, and discrimination based on race is manifested in various forms and levels [20]. We captured four types of discriminatory experiences reported by Chinese American parents and youth that were due to COVID-19 [20,21]. Individuals can be the *direct* targets of racial discrimination, but also *vicariously* witness, read, or hear about an incident of discrimination directed at same-race others [22]. Increases in virtual work and schooling from 2020 to 2021 have led to greater opportunities for *online*, in addition to *in-person*, experiences with discrimination [23]. Therefore, we examined both parents’ and youth’s experiences of *direct* and *indirect* (vicarious) racial discrimination, both *in person* and *online*.

To capture cultural-symbolic and sociopolitical manifestations of racism, we also assessed racism at the collective/group level. Specifically, we assessed two forms of Sinophobia: health-related Sinophobia, in which Americans consider the Chinese to be a health threat to American society, and media Sinophobia, which refers to the U.S. media’s perpetuation of Sinophobia during the pandemic [7]. Use of terms like the “Chinese virus” and “Wuhan virus” and the blaming of China for the pandemic by political leaders and the media likely fueled Sinophobia and online hate towards Chinese heritage individuals globally [24]. Finally, we examined Chinese American parents’ and youth’s levels of fear or worry that the previous administration’s use of the terms “China virus” and “Wuhan virus” in 2020 [25] would cause difficulties and safety concerns for Chinese Americans.

To summarize, we compared the rates of the following COVID-19 racism-related discrimination experiences, fear/worries, and mental health indices of Chinese American parents and youth in the Spring of 2020 and 2021: (1) online direct, (2) online vicarious, (3) in-person direct, and (4) in-person vicarious. We also assessed their perceptions of: (5) health-related Sinophobia, (6) Sinophobia in the media, and their (7) fear/worries about government-caused racism. In addition, we assessed parents’ and youth’s (8) psychological well-being and (9) generalized anxiety symptoms, and parents’ (10) depressive symptoms and youth’s (11) internalizing and externalizing problems. Finally, we assessed the associations between each of these racism-related experiences and mental health indices and compared the strength of these associations between 2020 and 2021.

## 2. Materials and Methods

Parents who self-identified as ethnically Chinese residing in the United States with at least one 4- to 18-year-old child were eligible to participate. The 2020 data were collected between March to May 2020 and included 529 Chinese American parents [7]. From this convenience sample, the 10- to 18-year-old children completed their own survey, resulting in a subsample of 225 parent–child dyads. The 2021 data were collected between January to April 2021, which included 515 Chinese American parents and a subsample of 316 parent–child dyads. Further details related to the two samples can be found in Appendix A.

Participants were recruited through phone calls and distribution of flyers via e-mails to different Chinese American organizations, and Facebook and WeChat groups. This recruitment strategy was necessary because we targeted a specific minority group within a short period, and social distancing regulations did not allow for community-based data collection. Parents provided consent for themselves and their children. Youth’s online assent was obtained separately. The surveys were hosted on the *Qualtrics* online platform. Rigorous data legitimacy checks (e.g., missing rate, duration, patterns of completion, self-contradictory answers) were conducted to ensure the validity and accuracy of the data [26]. Measures were available in English, simplified, or traditional Chinese using the back-translation method [27]. Parents and youth received e-gift cards as compensation.

### 2.1. Measures

Parents reported on their family’s demographic characteristics. Parents and youth separately reported on their experiences with four types of COVID-19-related racial discrimination, perceptions of health-related and media Sinophobia, government-related fear/worries, and their mental health. Details on the measures of different types of discrimination, Sinophobia, and government-related worries can be found in Appendix A.

*Online direct racial discrimination* was assessed using four items (e.g., “Due to COVID-19, people have said mean or rude things about me because of my race or ethnic group online”; parents α = 0.87, youth α = 0.88 in 2020; parents α = 0.83, youth α = 0.86 in 2021) [7]. *Online vicarious racial discrimination* was assessed using three items (e.g., “Due to COVID-19, people have cracked jokes about people of my race or ethnic group online”; parents α = 0.89, youth α = 0.89 in 2020; parents α = 0.86, youth α = 0.84 in 2021). Respondents rated how often they experienced each incident on a 5-point scale ranging from 1 (never) to 5 (every day).

*In-person direct racial discrimination* was assessed using five items for the parent survey and four items were used for the youth survey (e.g., “Some people were unfriendly or unwelcoming toward me because of my Chinese background”; parents α = 0.94, youth α = 0.83 in 2020; parents α = 0.98, youth α = 0.98 in 2021) [7]. Respondents rated how often they experienced each incident because of COVID-19 on a 5-point scale ranging from 1 (never) to 5 (every day).

*In-person vicarious racial discrimination* was measured using four items (e.g., “Someone said something negative about Chinese people [for example, their diet] related to the COVID-19 pandemic”; parents α = 0.86, youth α = 0.84 in 2020; parents α = 0.92, youth α = 0.92 in 2021) [7]. Respondents rated how often they experienced each incident on a 5-point scale ranging from 1 (never) to 5 (every day).

*Health-related Sinophobia* was assessed using three items (e.g., “A lot of Americans consider Chinese people as a threat to public health in America”; parents α = 0.70, youth α = 0.66 in 2020; parents α = 0.89, youth α = 0.95 in 2021) [7]. *Sinophobia in the media* was assessed using five items (e.g., “U.S. media presents Chinese people as dangerous”; parents α = 0.95, youth α = 0.89 in 2020; parents α = 0.96, youth α = 0.96 in 2021) [7]. Respondents indicated their agreement with each item using a 5-point scale ranging from 1 (strongly disagree) to 5 (strongly agree).

*Government-related fear/worries* was measured using three items that were created for the purpose of this study to capture Chinese American parents’ and youth’s fear, worries, and concerns for their safety and well-being as racial minorities due to the U.S. government’s handling and responses to the COVID-19 pandemic (e.g., “I am concerned that the government’s response to the COVID-19 outbreak [e.g., blaming China, calling it the “China virus” or “Wuhan Virus”] will make life more difficult for Chinese Americans; ” parents α = 0.91, youth α = 0.83 in 2020; parents α = 0.89, youth α = 0.87 in 2021). Items were rated on a 4-point scale ranging from 1 (strongly disagree) to 4 (strongly agree).

*Psychological well-being* was measured using Ryff’s 18-item Psychological Well-Being Scale [28]. Respondents rated the items (e.g., “When I look at the story of my life, I am pleased with how things have turned out”; parents α = 0.86, youth α = 0.83 in 2020; parents α = 0.87, youth α = 0.83 in 2021) on a 7-point scale ranging from 1 (strongly disagree) to 7 (strongly agree).

*Generalized anxiety symptoms* were assessed using the 7-item Generalized Anxiety Disorder Screener [29]. Respondents rated how often they had been bothered by each symptom over the past 2 weeks (parents α = 0.94, youth α = 0.89 in 2020; parents α = 0.94, youth α = 0.91 in 2021) on a 4-point scale ranging from 0 (not at all) to 3 (nearly every day).

*Depressive symptoms* in parents were measured using the 21-item Beck Depression Inventory-II [30]. Only parents rated the presence and severity of their depressive symptoms during the past 2 weeks (α = 0.93 in 2020 and α = 0.93 in 2021) on a 4-point scale from 0 to 3.

*Internalizing and externalizing problems* in youth were assessed using the Strengths and Difficulties Questionnaire (SDQ) [31]. Only youth rated 20 items assessing internalizing (emotional symptoms and peer relationships) and externalizing (hyperactivity and conduct) difficulties during the COVID-19 pandemic (internalizing problems α = 0.72, externalizing problems α = 0.76 in 2020; internalizing problems α = 0.74, externalizing problems α = 0.77 in 2021) on a 3-point scale ranging from 0 (not true) to 2 (certainly true).

### 2.2. Analyses

In 2020, 3% and 2% of the data were missing in the parent and the dyad sample, respectively. In 2021, 0.5% and 2% of the data were missing in the parent and the dyad sample, respectively. Missing patterns in all samples were completely at random and were handled using the maximum likelihood approach. The data were transformed to a long format to include a Year of Data Collection variable (i.e., 2020 versus 2021) and a Participation Status variable (i.e., whether a participant completed the survey in one or both years) for the analyses.

First, a set of ANCOVA analyses were conducted to compare parents’ and youth’s experiences of the four types of COVID-19-related racial discrimination, perceived health-related Sinophobia and media Sinophobia, government-related fear/worries, and mental health indices between 2020 and 2021. The year of data collection was the independent variable and the variables listed above were the outcomes in each of the ANCOVA models. Participation status was entered as a covariate in each model. When comparing parent variables, parent age, gender, nativity, education, household SES, and region of residence were also controlled as covariates. When comparing youth variables, youth age, gender, nativity, household SES, and region of residence were controlled as covariates. SES was coded using the Hollingshead Four-Factor Index of Social Status-Revised [32].

Next, three sets of moderated multiple regression models were conducted to compare the following associations across 2020 and 2021: (1) parents’ racial discrimination experiences and their own mental health; (2) parents’ racial discrimination experiences and their children’s mental health; and (3) youth’s racial discrimination experiences and their own mental health. In each model, one type of racial discrimination, Sinophobia perception, or government-related fear/worries, the data collection year, and the interaction between the discrimination variable and the data collection year were examined as predictors, and each mental health outcome was examined as the outcome variable. Covariates included parents’ and youth’s participation status, age, gender, nativity, household SES, and region of residence. Simple slope analyses were conducted to probe significant interactions between discrimination and Year variables in predicting mental health outcomes. Confidence intervals were obtained using the bootstrapping method of 2000 resampling replications.

## 3. Results

Table 1 and Table 2 present demographic characteristics and descriptive statistics for all study variables for the parent sample and the dyad sample, respectively. Parent participants were mostly mothers (T1: 80%, T2: 79%), foreign-born (T1: 98%, T2: 97%), residing in the southern region of the United States (T1: 76%, T2: 55%), well-educated/having a Bachelor’s degree or higher (T1: 87%, T2: 76%), and employed (T1: 89%, T2: 85%). On average, parents were in their 40s and had lived in the United States for more than 16 years. Most youth were U.S.-born.

### 3.1. Comparisons between 2020 and 2021 in Levels of Discrimination, Sinophobia, Government-Related Fear/Worries, and Mental Health Indices

Table 3 presents the results of the ANCOVA models. After controlling for participation status and demographic covariates, compared to 2020, parents in 2021 reported higher levels of: (1) online direct discrimination (EM_mean_ = −0.28, *p* < 0.001, ŋ_p_^2^ = 0.03), (2) in-person direct discrimination (EM_mean_ = −0.34, *p* < 0.001, ŋ_p_^2^ = 0.03), and (3) perceived media Sinophobia (EM_mean_ = −0.37, *p* < 0.001, ŋ_p_^2^ = 0.03), and lower levels of: (1) in-person vicarious discrimination (EM_mean_ = 0.34, *p* < 0.001, ŋ_p_^2^ = 0.02), and (2) psychological well-being (EM_mean_ = 0.11, *p* < 0.05, ŋ_p_^2^ = 0.01). Similarly, youth in 2021 reported higher levels of: (1) online direct discrimination (EM_mean_ = −0.22, *p* < 0.05, ŋ_p_^2^ = 0.01), (2) in-person direct discrimination (EM_mean_ = −0.43, *p* < 0.001, ŋ_p_^2^ = 0.04), but also (3) generalized anxiety (EM_mean_ = −1.14, *p* < 0.01, ŋ_p_^2^ = 0.02), and significantly lower levels of: (1) in-person vicarious discrimination (EM_mean_ = 0.54, *p* < 0.001, ŋ_p_^2^ = 0.06), (2) perceived health Sinophobia (EM_mean_ = −0.39, *p* < 0.001, ŋ_p_^2^ = 0.04), and (3) psychological well-being (EM_mean_ = 0.20, *p* < 0.01, ŋ_p_^2^ = 0.02) than in 2020 (Figure 1). No significant differences were found for other parent- or child-reported variables across the two years.

We also compared parents’ and youth’s rates of reporting slightly elevated or substantial risks of clinically significant mental health problems or difficulties (see cutoffs in Appendix A). For anxiety symptoms, 10% of parents reported a slightly elevated risk and 8% of parents reported a substantial risk, which did not differ from the rates in 2020 (9% and 5%, respectively). In contrast, more youth reported a slightly elevated risk of anxiety symptoms in 2021 (18%) than in 2020 (8%), whereas the percentage of youth reporting a substantial risk did not differ (6% in 2021 and 3% in 2020). For depressive symptoms, 24% of parents reported a slightly elevated risk, which was significantly higher than the percentage in 2020 (15%), whereas the percentage reporting a substantial risk of developing depressive symptoms did not differ across the two years (3% in 2021 and 4% in 2020). For internalizing and externalizing difficulties, more youth reported a substantial risk in 2021 (12%) than in 2020 (6%), whereas the rates of reporting slightly elevated risks did not differ (26% in 2021 and 23% in 2020).

### 3.2. Comparisons between 2020 and 2021 in the Associations between Racial Discrimination and Mental Health Indices

*Parent-parent associations*. Table 4 presents the results of the regression models. All indices of racism were significantly associated with poorer mental health. For parents, the associations between online vicarious discrimination, in-person vicarious discrimination, health-related and media Sinophobia perceptions, and government-related fear/worries and parent mental health outcomes were greater in 2021 compared to 2020. Specifically, parent online vicarious discrimination, in-person vicarious discrimination, and health-related and media Sinophobia perceptions were more strongly associated with poorer parent psychological well-being (interaction between the discrimination variable and the data collection year *b*s ranging from −0.19 to −0.09, *p*s ranging from 0.041 to <0.001) and more strongly positively associated with parent anxiety symptoms (interaction effect *b*s ranging from 0.74 to 1.76, *p*s ranging from 0.004 to <0.001) in 2021 than in 2020.

In addition, parent online vicarious discrimination was more strongly and positively associated with parent depressive symptoms (interaction effect *b* = 1.03, *p* < 0.001), and parent government-related worries were more strongly negatively associated with parent psychological well-being (interaction effect *b* = −0.11, *p* = 0.047) in 2021 than in 2020. In contrast, parent online direct discrimination experiences were less strongly positively associated with parent depressive symptoms (interaction effect *b* = –2.39, *p* < 0.001) and parent in-person direct discrimination was less strongly positively associated with parent depressive symptoms (interaction effect *b* = −1.71, *p* < 0.001) and less strongly negatively associated with parent psychological well-being (interaction effect *b* = −0.12, *p* = 0.004) in 2021 than in 2020.

*Youth-youth associations.* For the associations between youth discrimination variables and youth mental health outcomes, a similar pattern was observed, such that youth online vicarious discrimination, in-person vicarious discrimination, and health-related and media Sinophobia perceptions were more strongly positively associated with youth anxiety symptoms (interaction effect *b*s ranging from 0.92 to 1.65, *p*s ranging from 0.013 to <0.001) and internalizing problems (interaction effect *b*s ranging from 0.63 to 1.14, *p*s ranging from 0.022 to <0.001) in 2021 than in 2020. In contrast, the magnitude of the associations between youth online direct discrimination, in-person direct discrimination, and government worries and the four youth mental health outcomes did not differ (interaction effect *b*s ranging from −0.57 to 0.93, all *p*s > 0.05) between the two years, although all remained significant.

*Parent-youth associations.* Further, the associations between parent discrimination variables and youth mental health outcomes were also stronger in 2021 compared to 2020. Specifically, parent online vicarious discrimination, in-person vicarious discrimination, and health-related and media Sinophobia perceptions were more strongly negatively associated with youth psychological well-being (interaction effect *b*s ranging from −0.30 to −0.20, *p*s ranging from 0.002 to <0.001), more strongly positively associated with youth anxiety symptoms (interaction effect *b*s ranging from 1.17 to 1.39, *p*s ranging from 0.002 to <0.001), and more strongly positively associated with youth internalizing and externalizing problems (interaction effect *b*s ranging from 0.65 to 1.15, *p*s ranging from 0.030 to <0.001), except for the non-significant interaction terms for the associations between parent health-related and media Sinophobia and youth externalizing problems, in 2021 than in 2020. In addition, parent government-related worries were more strongly negatively associated with youth psychological well-being (interaction effect *b* = −0.22, *p* = 0.015) in 2021 than in 2020. In contrast, no significant differences were observed for the associations between parent online and in-person direct discrimination and the four youth mental health outcomes (interaction effect *b*s ranging from −0.83 to 0.96, all *p*s > 0.05) in 2021 compared to 2020, although all remained significant.

## 4. Discussion

Our findings revealed that a high percentage of both Chinese American parents and their children personally experienced or witnessed anti-Chinese/Asian racial discrimination both online and in-person due to the COVID-19 pandemic in 2021. The rates of vicarious discrimination decreased for in-person experiences and did not change significantly for online experiences between 2020 and 2021. However, these numbers were already high in 2020 and remained so in 2021, with 14–18% of parents and youth reporting vicarious racial discrimination almost every day and about 80% of parents and youth reporting experiencing or witnessing racial discrimination targeting Asian individuals due to COVID-19 at least once in 2021.

Importantly, the experiences of *direct* victimization both online and in person were significantly higher in 2021 compared to 2020. The increase in experiences of direct discrimination at least once is alarming (51% for parents and 50% for youth in 2020 to 55% for parents and 58% for youth in 2021) and may reflect more people returning to in-person settings for work, school, and public spaces in 2021, which increases the risk of racism-related exposure [33]. These numbers are consistent with other reports showing Asian Americans experiencing high rates of discrimination. For example, one study found that 58.7% of their Asian American participants experienced direct online racism and 88.1% experienced vicarious racism in 2020 [34]. Another national study found that 60.7% of Asian Americans and 64.7% Chinese Americans reported at least one incident of discrimination experiences during the COVID-19 pandemic, based on data collected across 2020 and 2021 [35].

In examining perceptions of collective racism towards their racial-ethnic group (Sinophobia), both parents’ and youth’s Sinophobia perceptions did not vary significantly across two years. However, parents were more likely to perceive that Sinophobia against Chinese Americans was presented and promoted by the media in 2021 than in 2020, indicating continued and manifested negative Sinophobic messaging over time. Despite the change in administration, in 2021, Chinese American parents’ and youth’s heightened worries, fear, and concerns for their safety and well-being as racial minorities due to the Trump administration’s handling and responses to the COVID-19 pandemic did not decrease significantly from 2020. These findings support Tong et al.’s [36] national survey results that Asian Americans perceived high levels of Asian hate online (a mean score of 3.72 on a 5-point scale) in 2020 and McGarity-Palmer et al.’s [37] study that Chinese Americans rated a mean score of 3.75 on a 5-point scale measuring collective racism/Sinophobia in 2021.

Regarding mental health indices, both parents and youth reported poorer mental health in 2021 than in 2020. These data indicated that more parents reported slightly elevated risks for developing depressive symptoms (24% versus 15%) and more youth reported slightly elevated risks for developing anxiety symptoms (18% versus 8%) in 2021 compared to 2020. These elevated mental health risk percentage numbers in 2021 were also higher than population norms (15% versus 13%, respectively for depressive symptoms and 5% versus 4%, respectively, for anxiety symptoms) [38,39], suggesting a cumulative effect of these racial stressors on Chinese American families’ mental health and well-being. Further, Chinese American youth also reported a higher percentage of substantial risk for general adjustment difficulties (6% versus 12%) in 2021 than in 2020. These trends align with the survey results that 46.0% of Asian Americans reported anxiety and about 30.0% of Asian American youth reported clinically elevated depressive symptoms and anxiety in 2020, and 48.7% of Asian Americans reported depressive symptoms and anxiety in 2021 [18].

In 2021, all types of COVID-19-related racial discrimination and racism were associated with poorer mental health among Chinese American parents and youth, consistent with previous studies on non-COVID-19-related discrimination [40]. Further, although the rates of direct experiences of racial discrimination for both parents and youth (online and in-person) were significantly higher in 2021 than 2020, direct experiences of racial discrimination were less strongly associated with higher depressive symptoms and poorer psychological well-being among parents (but not youth) in 2021 compared to 2020. Importantly, the positive associations remained significant in 2021, and there was no significant decrease in the strengths of association among youth, indicating that youth are more vulnerable to direct racial victimization than their parents in 2021 likely due to their developing social-cognitive and identity processes [41].

In contrast, vicarious discrimination experiences either online or in-person were generally associated more strongly with negative mental health indices in 2021 compared to 2020 for both parents and youth. These findings highlight the detrimental long-term impact of vicarious discrimination, where injustices committed against other members of the same social group are collectively shared and can be personally stressful for Chinese American families [22,42]. This result is especially concerning because videos and news reports on violent anti-Asian acts are widely circulated on social media [43]. Thus, we call for more attention on vicarious experiences of discrimination and potential ways to intervene in order to reduce its negative mental health impact.

Regarding collective racism, both health-related and media Sinophobia were more strongly associated with poor mental health in 2021 than in 2020. Moreover, Chinese American parents’ fears and worries about being targeted and having more difficult lives due to the Trump administration’s racist responses to the COVID-19 pandemic were more strongly associated with their poorer psychological well-being in 2021 compared to 2020, although the mean levels were not significantly different across the years. Together, these findings imply that vicarious experiences with discrimination, collective racism, and institutional racism may have a stronger negative toll on these families’ mental health further along in the pandemic, perhaps because they are perceived to be more systemic and widespread, and thus, less controllable or changeable [44].

When we examined Chinese American parents’ experiences of racism in relation to their children’s self-reported mental health, a similar pattern was found where parents’ vicarious (but not direct) experiences of racial discrimination both online and in-person and parents’ Sinophobia perceptions were more strongly associated with *all* indices of poorer mental health in youth. Chinese American parents’ government-related worries and concerns were further associated with their youth’s poorer psychological well-being. Parents’ own racial victimization experiences and feelings of being unsafe might impact their children’s mental health either directly or indirectly through a more stressful or hostile family environment, and/or maladaptive parenting [45]. Given the recent attention on anti-Asian hate crimes, mental healthcare professionals may be more focused on adults’ experiences and/or youth’s direct experiences with racial discrimination. However, our findings indicate clearly that attention must also be paid to addressing youth’s vicarious experiences with racism and concerns about collective and institutional racism beyond the pandemic [46,47]. In addition, how Chinese American youth might be impacted by processes within the family system stemming from racism needs further attention [45].

Although the course of the pandemic is waning, pediatricians and other healthcare professionals should continue to be sensitive to the mental health needs of Chinese American youth and their parents related to various forms of racism at different levels. The cumulative negative effects of racism, particularly vicarious discrimination and collective forms of racism may continue to have adverse health effects on Chinese Americans. We recommend that healthcare providers continue to identify youth with mental health concerns and to facilitate referrals to mental health services using culturally competent approaches that validate and normalize mental health help-seeking behavior among Asian Americans.

### 4.1. Limitations

Although our sample comprised families from 30 states, it was not nationally representative of Chinese Americans due to the urgent need for rapid data collection. Thus, we were unable to examine regional differences in discrimination experiences and their correlates [40]. In addition, statistical control for social desirability bias and the inclusion of different reporters are recommended for future research to address the issues of using self-reported data.

Further, no causal interpretations can be made as our data were cross-sectional, although longitudinal studies have found that discrimination predicts mental health and not the other way [48,49]. In addition, our sample was generally well-educated and our results may not be generalizable to families from lower educational backgrounds. Finally, we focused on the experiences of Chinese Americans, but the study of shared and unique experiences and effects of racism in other Asian American subgroups is necessary [50].

### 4.2. Implications for Policy and Practice

Governments and health authorities should denounce inflammatory political rhetoric that contributes to the scapegoating of Asian Americans for the public health crisis (e.g., discriminatory languages such as terms like the “China virus”) [12,51]. The government should also discourage hate speech in social media that perpetuates Sinophobia and racism, which contributes to the mental health struggles of Asian Americans. Clear rules and regulations against hate crime and harassment should be established (e.g., passing bills to address public safety issues on public transit systems) and information on legal channels for victims to report discrimination/racism along with mental health resources related to discrimination should be widely disseminated [51]. In addition, the government should increase funding for Asian American community organizations to support victims and for cultural competency, implicit bias, and bystander training for educators and healthcare providers to support youth and families [51,52].

## 5. Conclusions

During the COVID-19 pandemic, the salience of anti-Asian sentiments and violence targeting Asian Americans increased and led to several key medical commentaries [49,53] and social policy initiatives [54]. As social distancing regulations relaxed and greater intergroup interactions occurred in 2021, various sources and datasets documented increases in acts of anti-Asian racism [2,33,55], as was also documented in this current study. Our findings contribute to this literature by including both Chinese American parents and their children across two years. High rates of racial discrimination across multiple dimensions and the detrimental impacts on Chinese American families’ mental health and well-being were found in the second year of the pandemic. We also documented that for Chinese American parents and their children, vicarious and collective racism may have even stronger negative impacts on their mental health and well-being later in the pandemic (in 2021). While almost all indices of racism were significantly associated with poorer mental health, as the negative impact of vicarious and collective racism accumulated over time (from 2020 to 2021), they appeared to take a stronger toll on Chinese American youth’s and parents’ mental health as shown by the stronger relation between these types of racism and mental health in 2021 compared with 2020. Parents’ experience of racism also had a stronger trickle down (downstream) effect on youth mental health.

Future research identifying protective factors to decrease and ameliorate the negative effects of these experiences is imperative [4,10,14]. Our findings call for continued effective public health and educational strategies to decrease the stigmatization of and discrimination against Asian Americans [56], increased attention to their mental health needs related to racism, particularly among pediatric populations, and the provision of culturally sensitive care addressing unique challenges of Asian American families. Importantly, racism is perpetuated at individual, cultural, institutional, and structural levels and has harmful psychological and physiological effects [57]. Thus, efforts to decrease health disparities in this growing population and other communities of color require extensive, long-term national commitments to eliminating structural aspects of racism [57].

## Figures and Tables

**Figure 1 ijerph-20-05437-f001:**
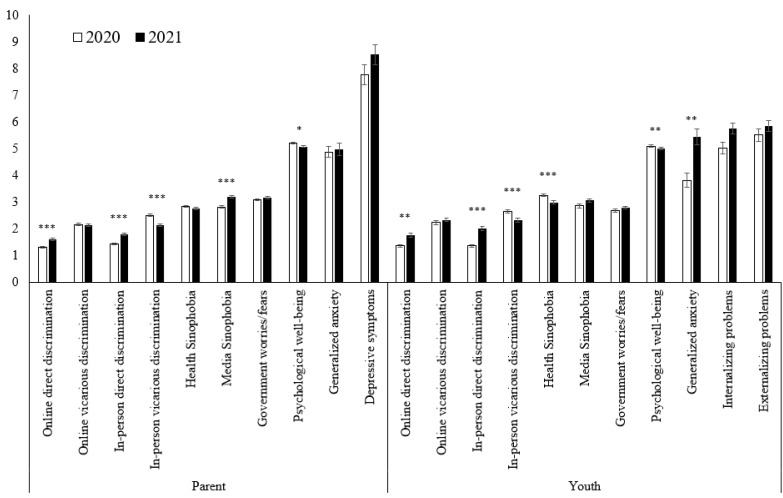
Comparisons of parents’ and youth’s discrimination, Sinophobia perceptions, government-related worries, and mental health indices between 2020 and 2021 from ANCOVA models. Note. Means presented in this figure are observed values and the mean differences between observed values may be different from estimated marginal mean differences that controlled for covariates in Table 3. The statistical test of the mean differences is presented based on the ANCOVA models. * *p* < 0.05, ** *p* < 0.01, *** *p* < 0.001.

**Table 1 ijerph-20-05437-t001:** Statistics of demographic characteristics and variables of interest in the parent samples in 2020 and 2021.

	2020 Parent Sample (*N* = 529)		2021 Parent Sample (*N* = 515)	
	Parent	Child	Parent	Child
Demographic Characteristics				
Age mean (*SD*)	43.42 (6.48)	11.71 (4.02)	42.88 (6.72)	12.52 (3.76)
Age range	28–64	4–18	28–74	4–19
Years in the U.S. Mean (*SD*)	16.73 (9.69)	10.18 (4.70)	16.77 (8.54)	11.54 (4.40)
Nativity				
Foreign born	508 (96%)	123 (23%)	503 (98%)	96 (19%)
U.S. born	21 (4%)	406 (77%)	12 (2%)	419 (81%)
Parent education				
Less than high school graduate	39 (7%)	-	27 (5%)	-
High school graduate	37 (7%)	-	34 (7%)	-
Some college	25 (5%)	-	59 (12%)	-
College graduate	112 (21%)	-	119 (23%)	-
Graduate/Professional degree	316 (60%)	-	276 (54%)	-
Parent marital status				
Married or remarried	474 (90%)	-	467 (91%)	-
Divorced, separated, or widowed	55 (10%)	-	48 (9%)	-
Parent occupation				
Administrators, professionals, and large business owners	284 (54%)	-	260 (51%)	-
Technicians and small business owners	97 (18%)	-	93 (18%)	-
Skilled workers	51 (10%)	-	55 (11%)	-
Temporary workers	27 (5%)	-	20 (4%)	-
Housewife or unemployed	70 (13%)	-	86 (17%)	-
Parent reporter				
Mother	416 (79%)	-	404 (78%)	-
Father	113 (21%)	-	111 (22%)	-
Child gender				
Boys	-	280 (53%)	-	263 (51%)
Girls	-	249 (47%)	-	252 (49%)
Family region				
Northeast	94 (18%)		97 (19%)	
South	352 (67%)		290 (56%)	
Midwest	32 (6%)		33 (6%)	
West	51 (10%)		95 (18%)	
COVID-19 Racial Discrimination Mean (SD)				
Online direct	1.31 (0.62)	-	1.59 (1.13)	-
Online vicarious	2.15 (1.11)	-	2.12 (1.18)	-
In-person direct	1.42 (0.69)	-	1.78 (1.20)	-
In-person vicarious	2.49 (1.15)	-	2.12 (1.18)	-
Health Sinophobia	2.83 (0.94)	-	2.75 (1.07)	-
Media Sinophobia	2.81 (1.16)	-	3.18 (1.10)	-
Government worries/fears	3.08 (0.90)	-	3.16 (0.81)	-
Mental Health Mean (SD)				
Psychological well-being	5.19 (0.82)	-	5.07 (0.83)	-
Generalized anxiety	4.87 (4.76)	-	4.97 (5.37)	-
Depressive symptoms	7.76 (8.73)	-	8.51 (8.42)	-

Note: Online direct and vicarious discrimination, in-person direct and vicarious discrimination, and health and media Sinophobia were measured on 5-point scales (1–5); Government worries/fears were measured on a 4-point scale (1–4); Psychological well-being was measured on a 7-point scale (1–7); Generalized anxiety and depressive symptoms were measured on 4-point scales (0–3).

**Table 2 ijerph-20-05437-t002:** Statistics of demographic characteristics and variables of interest in the dyad samples in 2020 and 2021.

	2020 Dyad Sample (*N* = 225)		2021 Dyad Sample (*N* = 316)	
	Parent	Child	Parent	Child
Demographic Characteristics				
Age mean (*SD*)	46.07 (5.15)	13.84 (2.41)	44.13 (5.86)	14.12 (2.36)
Age range	28–64	10–18	28–64	10–19
Years in the U.S. Mean (*SD*)	18.59 (8.40)	12.42 (3.78)	17.86 (7.23)	13.20 (3.53)
Nativity				
Foreign born	220 (98%)	48 (21%)	307 (97%)	52 (16%)
U.S. born	5 (2%)	177 (79%)	9 (3%)	264 (84%)
Parent education				
Less than high school graduate	13 (6%)	-	13 (4%)	-
High school graduate	12 (5%)	-	21 (7%)	-
Some college	6 (3%)	-	42 (13%)	-
College graduate	51 (23%)	-	63 (20%)	-
Graduate/Professional degree	143 (64%)	-	177 (56%)	-
Parent marital status				
Married or remarried	466 (91%)	-	287 (91%)	-
Divorced, separated, or widowed	48 (9%)	-	29 (9%)	-
Parent occupation				
Administrators, professionals, and large business owners	139 (62%)	-	170 (54%)	-
Technicians and small business owners	31 (14%)	-	52 (17%)	-
Skilled workers	20 (9%)	-	36 (11%)	-
Temporary workers	10 (4%)	-	12 (4%)	-
Housewife or unemployed	25 (11%)	-	46 (15%)	-
Parent reporter				
Mother	180 (80%)	-	249 (79%)	-
Father	45 (20%)	-	67 (21%)	-
Child gender				
Boys	-	118 (52%)	-	155 (50%)
Girls	-	107 (48%)	-	157 (50%)
Family region				
Northeast	22 (10%)		46 (15%)	
South	172 (76%)		174 (55%)	
Midwest	14 (6%)		21 (7%)	
West	17 (8%)		75 (24%)	
COVID-19 Racial Discrimination Mean (SD)				
Online direct	1.20 (0.44)	1.35 (0.64)	1.77 (1.31)	1.76 (1.24)
Online vicarious	2.13 (1.02)	2.23 (1.20)	2.23 (1.31)	2.33 (1.29)
In-person direct	1.31 (0.53)	1.35 (0.58)	1.94 (1.38)	1.99 (1.40)
In-person vicarious	2.39 (1.11)	2.64 (1.11)	2.29 (1.30)	2.32 (1.27)
Health Sinophobia	2.74 (0.94)	3.25 (0.84)	2.86 (1.16)	2.98 (1.16)
Media Sinophobia	2.66 (1.13)	2.85 (1.05)	3.25 (1.13)	3.05 (1.13)
Government worries/fears	3.05 (0.92)	2.67 (0.84)	3.18 (0.80)	2.78 (0.88)
Mental Health Mean (SD)				
Psychological well-being	5.32 (0.83)	5.09 (0.86)	5.09 (0.86)	4.99 (0.77)
Generalized anxiety	4.51 (4.95)	3.81 (4.16)	5.15 (5.67)	5.44 (5.19)
Depressive symptoms	6.61 (8.73)	-	8.72 (8.65)	-
Internalizing problems	-	5.01 (3.27)	-	5.74 (3.64)
Externalizing problems	-	5.50 (3.35)	-	5.85 (3.51)

Note: Online direct and vicarious discrimination, in-person direct and vicarious discrimination, and health and media Sinophobia were measured on 5-point scales (1–5); Government worries/fears were measured on a 4-point scale (1–4); Psychological well-being was measured on a 7-point scale (1–7); Generalized anxiety and depressive symptoms were measured on 4-point scales (0–3); Internalizing and externalizing problems were measured on 3-point scales (0–2).

**Table 3 ijerph-20-05437-t003:** Comparisons of parents’ and youth’s discrimination, Sinophobia perceptions, government-related worries, and mental health indices between 2020 and 2021 from ANCOVA models.

Variables	*F*	Estimated Marginal Mean Differences	Partial ŋ^2^
Parent online direct	27.81 ***	−0.28	0.03
Parent online vicarious	0.32	0.04	0.000
Parent in-person direct	33.14 ***	−0.34	0.03
Parent in-person vicarious	24.84 ***	0.34	0.02
Parent health Sinophobia	1.80	0.08	0.002
Parent media Sinophobia	26.86 ***	−0.37	0.03
Parent government worries	2.38	−0.09	0.003
Parent psychological well-being	5.30 *	0.11	0.01
Parent generalized anxiety	0.16	−0.01	0.000
Parent depressive symptoms	1.53	−0.64	0.002
Youth online direct	7.16 **	−0.22	0.01
Youth online vicarious	0.31	0.06	0.000
Youth in-person direct	23.31 ***	−0.43	0.04
Youth in-person vicarious	31.20 ***	0.54	0.06
Youth health Sinophobia	20.01 ***	0.39	0.04
Youth media Sinophobia	1.16	−0.10	0.002
Youth government worries	0.08	−0.02	0.000
Youth psychological well-being	7.95 **	0.20	0.02
Youth generalized anxiety	7.55 **	−1.14	0.02
Youth internalizing problems	1.31	−0.35	0.003
Youth externalizing problems	0.03	−0.05	0.000

Note: Parent participation status, age, gender, nativity, education, household SES, and family region in the U.S. were controlled as covariates when comparing parent variables; Dyad participation status, youth age, gender, nativity, household SES, and family region in the U.S. were controlled as covariates when comparing youth variables. Estimated marginal mean differences were differences between the two years controlling for the covariates. Positive marginal mean differences represent higher levels in 2020 than 2021. Parent variable comparison *df* = (1, 1033); Youth variable comparison *df* = (1, 531). * *p* < 0.05, ** *p* < 0.01, *** *p* < 0.001.

**Table 4 ijerph-20-05437-t004:** Results of the moderated multiple regression models.

Predictors: Types of Racial Discrimination × Year	*b*s of the Interaction Term for Parent Sample			*b*s of the Interaction Term for Parent-Youth Sample			
	Parent Psychological Well-Being	Parent Anxiety Symptoms	Parent Depressive Symptoms	Youth Psychological Well-Being	Youth Anxiety Symptoms	Youth Internalizing Problems	Youth Externalizing Problems
Parent online direct × Year	0.03 [−0.25 ***, −0.25 ***]	0.63 [1.80 ***, 2.61 ***]	−2.39 *** [4.36 ***, 2.04 ***]	−0.24 [−0.07, −0.31 ***]	0.002 [1.70 *, 2.01 ***]	−0.34 [1.79 **, 1.49 ***]	0.64 [−0.35, 0.33]
Parent online vicarious × Year	−0.19 *** [0.04, −0.16 ***]	1.76 *** [0.71 ***, 2.59 ***]	1.03 *** [1.39 ***, 2.65 ***]	−0.20 ** [−0.02, −0.22 ***]	1.23 *** [0.32, 1.65 ***]	1.15 *** [0.23, 1.36 ***]	0.86 ** [−0.12, 0.75 ***]
Parent in-person direct × Year	0.11 * [−0.33 ***, −0.23 ***]	0.19 [2.38 ***, 2.66 ***]	−1.71 *** [4.08 ***, 2.54 ***]	0.01 [−0.32 **, −0.28 ***]	−0.83 [2.33 ***, 1.77 ***]	−0.02 [1.43 **, 1.39 ***]	0.96 [−0.58, 0.40]
Parent in-person vicarious × Year	−0.12 ** [−0.03, −0.17 ***]	1.36 *** [1.20 ***, 2.74 ***]	0.48 [2.09 ***, 2.87 ***]	−0.22 *** [−0.02, −0.24 ***]	1.29 *** [0.40, 2.11 ***]	1.12 *** [0.35, 1.58 ***]	0.65 * [−0.21, 0.49 *]
Parent health Sinophobia × Year	−0.16 *** [−0.11 **, −0.29 ***]	1.37 *** [1.22 ***, 2.63 ***]	0.79 [1.66 ***, 2.63 ***]	−0.30 *** [−0.07, −0.38 ***]	1.39 *** [0.42, 1.91 ***]	1.12 *** [0.26, 1.31 ***]	0.39 [0.36, 0.77 ***]
Parent media Sinophobia × Year	−0.09 * [−0.09 **, −0.18 ***]	0.74 ** [1.16 ***, 1.81 ***]	0.18 [1.73 ***, 1.87 ***]	−0.25 *** [−0.03, −0.27 ***]	1.17 ** [0.29, 1.39 ***]	0.66 * [0.49 *, 1.06 ***]	0.14 [0.44 *, 0.57 **]
Parent government worries/fears × Year	−0.11 * [−0.05, −0.15 ***]	0.53 [1.33 ***, 1.71 ***]	0.27 [2.02 ***, 2.13 ***]	−0.22 * [0.01, −0.21 ***]	0.26 [0.71 *, 1.15 **]	0.46 [0.37, 0.92 ***]	0.38 [−0.01, 0.45]
Youth online direct × Year				−0.01 [−0.24 **, −0.24 ***]	0.31 [2.07 ***, 2.50 ***]	0.22 [1.53 ***, 1.69 ***]	0.19 [0.58, 0.87 ***]
Youth online vicarious × Year				−0.08 [−0.08, −0.14 ***]	1.65 *** [0.69 **, 2.17 ***]	1.06 *** [0.41 *, 1.33 ***]	0.60 * [0.04, 0.74 ***]
Youth in-person direct × Year				0.11 [−0.31 ***, −0.19 ***]	−0.37 [2.34 ***, 2.18 ***]	−0.57 [2.08 ***, 1.54 ***]	0.13 [0.48, 0.71 ***]
Youth in-person vicarious × Year				−0.10 [−0.06, −0.14 **]	1.27 *** [1.21 ***, 2.55 ***]	1.14 *** [0.56 **, 1.65 ***]	0.44 [0.33, 0.82 ***]
Youth health Sinophobia × Year				−0.07 [−0.14 *, −0.20 ***]	1.03 * [1.20 ***, 1.90 ***]	0.69 * [0.85 **, 1.26 ***]	−0.27 [1.10 ***, 0.73 ***]
Youth media Sinophobia × Year				−0.03 [−0.13 *, −0.15 ***]	0.92 ** [1.03 ***, 1.72 ***]	0.63 * [0.75 ***, 1.19 ***]	0.02 [0.77 ***, 0.78 ***]
Youth government worries/fears × Year				−0.11 [−0.12, −0.21 ***]	0.93 [1.47 ***, 2.11 ***]	0.47 [1.05 ***, 1.30 ***]	0.05 [0.66 *, 0.62 *]

Note: * *p* < 0.05. ** *p* < 0.01. *** *p* < 0.001. The numbers in the cells outside of the brackets represent the unstandardized regression coefficients for the interaction terms in the models. The numbers on the left within the brackets represent the unstandardized regression coefficients for the association between the discrimination variable and the outcome in 2020 and the numbers on the right represent the association in 2021. Moderated multiple regression models were run separately for each pair of predictor and outcome. Unstandardized regression coefficients are reported. Parent age, gender, education, nativity, household SES, and family region were controlled as covariates when parent racial discrimination, Sinophobia perception, or government worries/fears were analyzed as the predictor and parent outcome was analyzed as the outcome variable in the models. Parent education, parent and youth age, gender, nativity, household SES, and family residence region were controlled as covariates when parent racial discrimination, Sinophobia perception, or government worries/fears were analyzed as the predictor and youth outcome was analyzed as the outcome variable in the models. Youth age, gender, nativity, household SES, and family region were controlled as covariates when youth racial discrimination, Sinophobia perception, or government worries/fears were analyzed as the predictor and youth outcome was analyzed as the outcome variable in the models.

## Data Availability

The data presented in this study are available on reasonable request from the corresponding author. The data are not publicly available due to privacy restriction.

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
