# Peer review of "COVID-19 Racism and Chinese American Families’ Mental Health: A Comparison between 2020 and 2021"

_ijerph, 2023, doi:10.3390/ijerph20085437_

Round 1
Reviewer 1 Report
Dear authors,
I would like to thank you for the opportunity to review this manuscript. These are my comments:
Introduction:
I have no major remarks on the Introduction. Still, I consider that lines 67-71 could be removed because the general content overlaps with the content of the lines 89-98.
Material and methods:
I have to note the clarity and the rigorous quantitative research in this study.
Therefore, I consider that there are too many described parameters. I suggest you to make some simplifications, like online direct/vicarious racial discrimination and in-person direct/vicarious discrimination. After my opinion, there are too many results and association that dilute the main ideas revealed by this study.
Results:
In the lines 209-213 you could make a more detailed description of the demographic characteristics from Table 1 (percentages).
Discussion:
This chapter must be improved by comparing data from this study with data from other data regarding racial discrimination, Sinophobia and mental health indices.
Conclusions:
I suggest to complete this chapter with data resulted from the multiple regression models (Table 4).
Reviewer 2 Report
The research tackles important topic of Sinophobia and racism related to the COVID-19. I find the paper interesting, important and timely. I believe that this research fills the gap in the literature and can be of interest to the readers of the Journal. I have only some minor comment:
The Introduction should contextualize Sinophobia and racism related to the COVID-19 by providing the reader with some basic information on the early news coverage of the outbreak that associated the virus with China in a manner that contributed to stigma. While it could describe how the outbreak of the COVID-19 led to global panic and the spread of false information that resulted in increase in prejudice, discrimination, and Sinophobia. It should also mention that also previous epidemics, including SARS, H1N1, MERS, Zika, Ebola, and COVID-19 resulted in stigma towards various social groups:
--Yuan, K., Huang, XL., Yan, W. et al. A systematic review and meta-analysis on the prevalence of stigma in infectious diseases, including COVID-19: a call to action. Mol Psychiatry 27, 19–33 (2022). https://doi.org/10.1038/s41380-021-01295-8
Additionally, The Authors could show how government leaders and political parties in various countries, including in the United States, United Kingdom, Italy, Spain, Greece, France, and Germany, have encouraged, both directly or indirectly, racism, or xenophobia by using anti-Chinese rhetoric.
Finally, it would be interesting to show how term like the “Chinese virus” and the “Wuhan virus” spread in social media affecting Sinophobia, i.e.:
--Sakki I, Castrén L. Dehumanization through humour and conspiracies in online hate towards Chinese people during the COVID-19 pandemic. British Journal of Social Psychology 61(1) https://doi.org/10.1111/bjso.12543.
Additionally, although the paper cites the most relevant research in the topic there are some other works that could be cited, i.e.:
--Hahm, H.C., Xavier Hall, C.D., Garcia, K.T. et al. Experiences of COVID-19-related anti-Asian discrimination and affective reactions in a multiple race sample of U.S. young adults. BMC Public Health 21, 1563 (2021). https://doi.org/10.1186/s12889-021-11559-1
-- Shang Z, Kim JY, Cheng SO. Discrimination experienced by Asian Canadian and Asian American health care workers during the COVID-19 pandemic: a qualitative study. CMAJ Open. 2021;9(4):E998-E1004. https://doi.org/10.9778/cmajo.20210090.
-- Gao Z. Sinophobia during the Covid-19 Pandemic: Identity, Belonging, and International Politics. Integr Psychol Behav Sci. 2022;56(2):472-490. https://doi.org/10.1007/s12124-021-09659-z.
-- Croucher SM, Nguyen T and Rahmani D (2020) Prejudice Toward Asian Americans in the Covid-19 Pandemic: The Effects of Social Media Use in the United States. Front. Commun. 5:39. https://doi.org/10.3389/fcomm.2020.00039.
Line 35: for non-American reader the AAPI abbreviation should be explained.
Reviewer 3 Report
The study compared the prevalence of COVID-19-related racism and its negative impact on the mental health of Chinese American parents and youth between 2020 and 2021.This study highlights the ongoing issue of racism against Chinese Americans and its negative impact on their psychological health and well-being. It is useful to improve their individual health outcomes and promote greater social justice. However, there are some revisions needed to be made to improve the quality of the manuscript. Specific suggestions are as follows:
1. Does the abstract include specific metrics?
2. A section such as "Implications for Policy and Practice" can be added to the Discussion section to analyze the significance of the research for practice.
3. Present the results graphically to provide an intuitive sense of differences and changes between years.
4. The family conditions in the current study are relatively good. Can it be added to families with general or even poor family conditions to analyze and compare if there are differences in family conditions?
5. Can causality analysis be done using structural equation model (SEM)?
6. There are some spelling mistakes, such as L14. The manuscript should be checked carefully before publication.
